# Influence of Corrosion on Dynamic Behavior of Pedestrian Steel Bridges—Case Study

Susana Barrios [1], Andrés Guzmán [1,*] and Albert Ortiz [2]

1   Department of Civil and Environmental Engineering, Universidad del Norte, Barranquilla 081007, Colombia
2   School of Civil and Geomatics Engineering, Universidad del Valle, Cali 760042, Colombia
*   Correspondence: faguzman@uninorte.edu.co; Tel.: +57-30-0428-4680

**Abstract:** Corrosion directly affects the structural stiffness of a steel element, reducing the thickness, thus inertia, due to the gradual deterioration of the material. Quickly identifying corrosion damage to the stiffness of a steel structure is a challenge in coastal environments since corrosion progresses rapidly, and traditional methods of inspection and diagnosis are time-consuming and costly. This is an important issue; therefore, characterization of the corrosion level represents a key element in making decisions regarding maintenance or structural integrity. This work estimates the relationship between the corrosion level in steel structures and their dynamic parameters using ambient vibration records. It comprises the characterization of the dynamic behavior and corrosion state of three full-scale pedestrian bridges with similar geometry, material, and structural configuration characteristics but with significant differences in the degree of deterioration. The structures were instrumented with piezoelectric sensors connected to a portable data acquisition system; the recorded information was analyzed with optimization algorithms in Python based on the power spectral density (PSD) of the vibrations of each bridge. The parameters obtained related to the degree of corrosion determine the incidence of the level of deterioration in the structural behavior, thus involving changes in its stiffness and mass.

**Keywords:** pedestrian bridge; corrosion; ambient vibrations; structural health; modal identification algorithms





## 1. Introduction

Corrosion is a permanent threat to steel structures since it causes impairment of functionality, economic losses, and failures that can lead to accidents, loss of life, and productivity. The threat is even more clear and visible in coastal areas since deterioration, economic losses, and reinvestment are evident in less time [1]. Faced with these conditions of exposure and rapid progress of corrosion, several questions arise: how much has the stiffness of the structure been affected, and is it necessary to dismantle the structure? For this reason, it is necessary to use technology and methods [2] to rapidly identify the current state of the structures (deterioration of their properties that affect their mechanic and dynamic behavior) associated with the degree of visible deterioration. These methods allow us to identify the impact on the stiffness of the structure through the evaluation of some indicators, such as dynamic parameters that are directly related to the variables of interest. This has been the focus of recent research that has studied the effects of damage or deterioration of structures by identifying changes in their dynamic properties. Most recent literature has focused on using low-cost technology for this purpose [3,4]. Identification of modal parameters is important for obtaining relevant information for risk assessment and structural health control [5], which opens the door to new methodologies that allow institutions to perform non-destructive testing and quickly obtain a diagnosis for risk management and the execution of preventive and corrective maintenance plans.

The proposed case study is based on pedestrian bridges with a Warren-type truss structural configuration. Bridges, despite being built in steel [6], usually lack preventive

maintenance in developing countries [7–9]. These structures are considered to integrate the constructive heritage of the past century and represent a high patrimonial value that needs to be preserved [10,11]. This research aims to relate the degree of corrosion with the dynamic properties of structures; to accomplish this goal, three steel pedestrian bridges were selected, each with a characteristic deterioration degree due to corrosion (Figure 1). The civil works of maintenance due to corrosion are expensive, and its identification is critical [12]. To this end, countries such as the USA have developed bridge inspection and maintenance manuals [13]. The manuals specified methods for corrosion identification as visual and optical testing (VT), radiographic testing (RT), electromagnetic testing (ET), ultrasonic testing (UT), liquid penetrant testing (PT), magnetic particle testing (MT), acoustic emission testing (AE), and infrared and thermal testing (IR) [14].

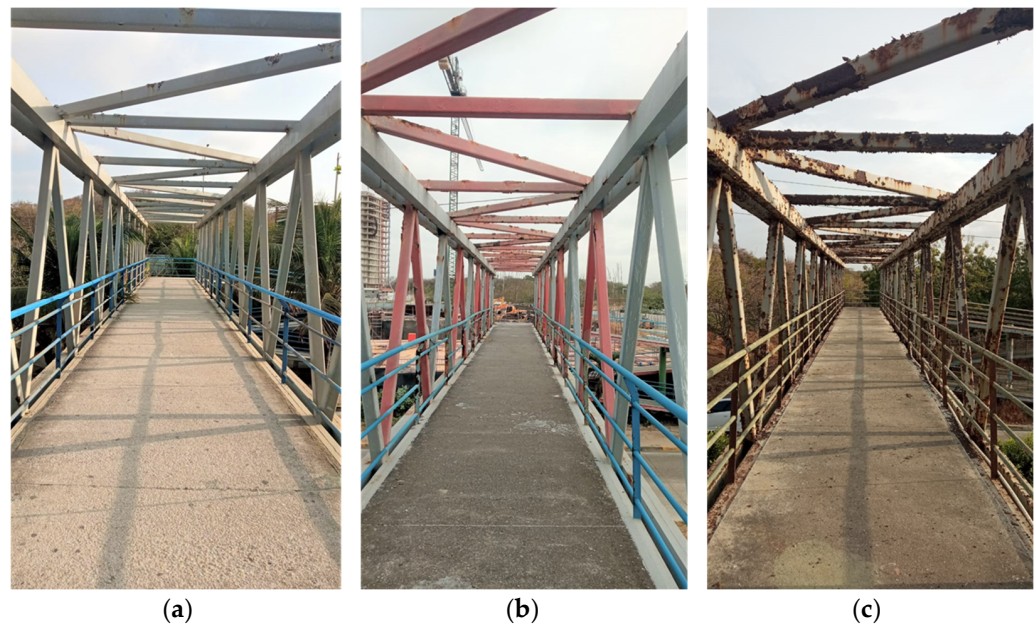

(a)         (b)         (c)

**Figure 1.** Current status of pedestrian bridges. (**a**) Uninorte, (**b**) Sagrado and, (**c**) Unilibre.

The bridges under evaluation are located in Puerto Colombia, Colombia, approximately 3.5 km from the Caribbean Sea (Figure 2a). The conditions at this location promote accelerated corrosion of the elements. The structures under study are similar in geometry, materials, and structural configuration, but differ significantly in their degree of deterioration and functionality (Figure 1). The difference in their degree of deterioration was caused by the different maintenance made by private institutions located nearby the bridges. Therefore, the question to solve is how the degree of corrosion of a pedestrian bridge affects the dynamic behavior of the structure. The bridges have been named after the educational institution next to them, and therefore, in this document, they will be referred to as "Uninorte" (Universidad del Norte), "Sagrado" (Sagrado Corazón School), and "Unilibre" (Universidad Libre). The linear distance between the Sagrado bridge and the Uninorte bridge is approximately 690 m, and the distance between the Uninorte bridge and the Unilibre bridge is approximately 1600 m, which indicates that the level of exposure to the marine environment is similar in the three bridges since the three structures are in a radius parallel to the spot described as Mallorquín swamp (lagoon) border Figure 2b.

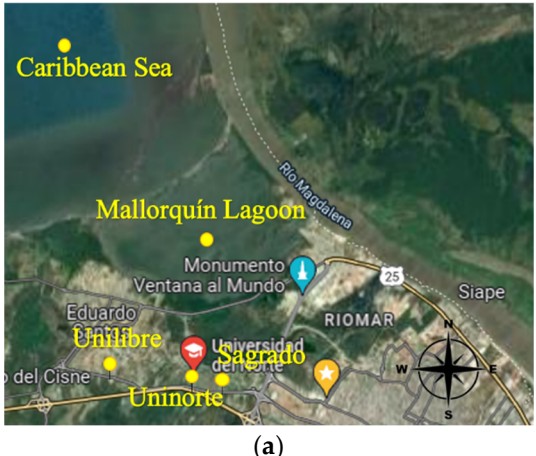

(**a**)

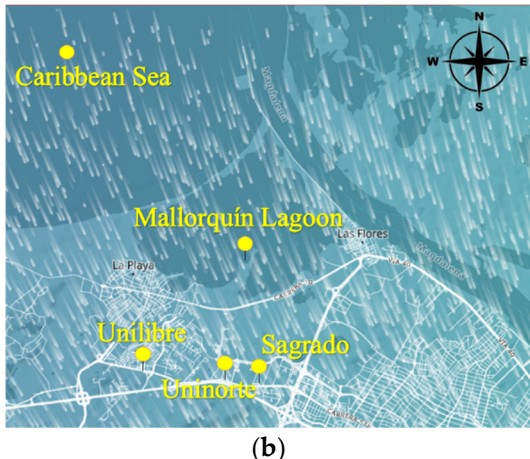

(**b**)

**Figure 2.** Geographic and environmental information. (**a**) Location of pedestrian bridges Uninorte, Sagrado, and Unilibre. (**b**) Wind direction (2022).

This research shows the results obtained for each structure when disturbed by ambient vibrations. Then, each bridge is dynamically characterized to identify its dynamic parameters and relate them to the degree of deterioration identified in the field. This Section describes the objective of this paper, the structures under study, and the location of the structures. Section 2 presents the background related to the modal parameters identification in the structures subjected to ambient vibrations and corrosion. It described the techniques and methods that have been developed for structural damage identification.

Section 3 describes the proposed methodology, which details three main branches that describe the work process. First, a geometric characterization is performed to comprehend the structures and describe them geometrically and analytically through mathematical models to have reference values. Then, in Section 3.2, the dynamic characterization of the structures is presented, which is the main focus of this study. This section described how the tests are carried out, the signals are processed, and experimental dynamic parameters are obtained.

Then, a characterization of the generalized corrosion degree of each bridge is performed quantitatively through the corrosion matrix and the deterioration observed in the field inspections. The dynamic parameters and the degree of corrosion are analyzed to validate if and to what extent they are statistically correlated. Section 4 presents the results obtained for each bridge and each branch or step shown in the methodology. Finally, Section 5 presents the discussions, conclusions, and future work for the research.

*Description of the Structures*

The Uninorte, Sagrado, and Unilibre bridges are steel pedestrian bridges with a Warren-type truss structural configuration. The bridge's geometry is similar to that shown in Figures 1 and 3–6, Table 1, and Section 4.1.1 (Bridges geometry).

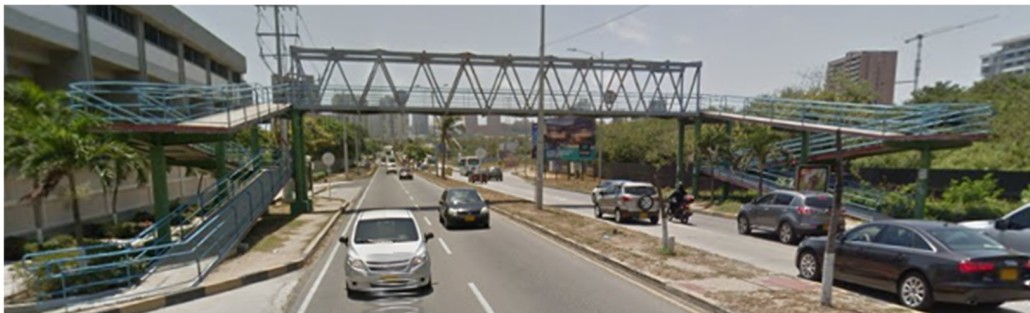

**Figure 3.** General view for the pedestrian bridges; Warren-type truss (Sagrado).

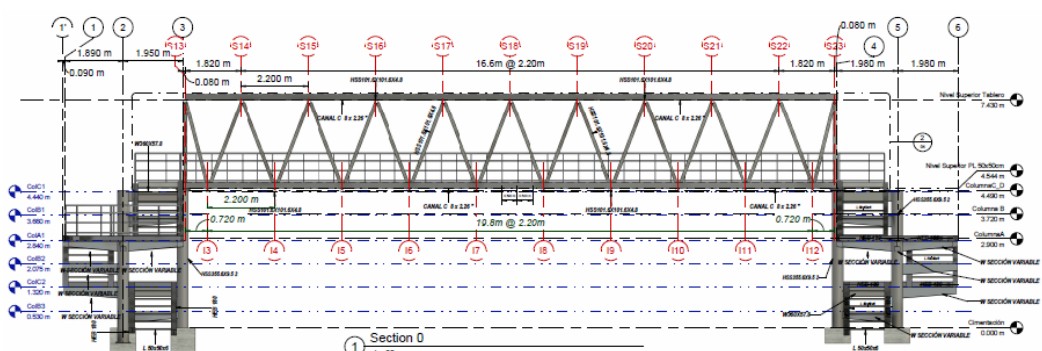

**Figure 4.** Lateral view of the design of the pedestrian bridges under study (schematic view).

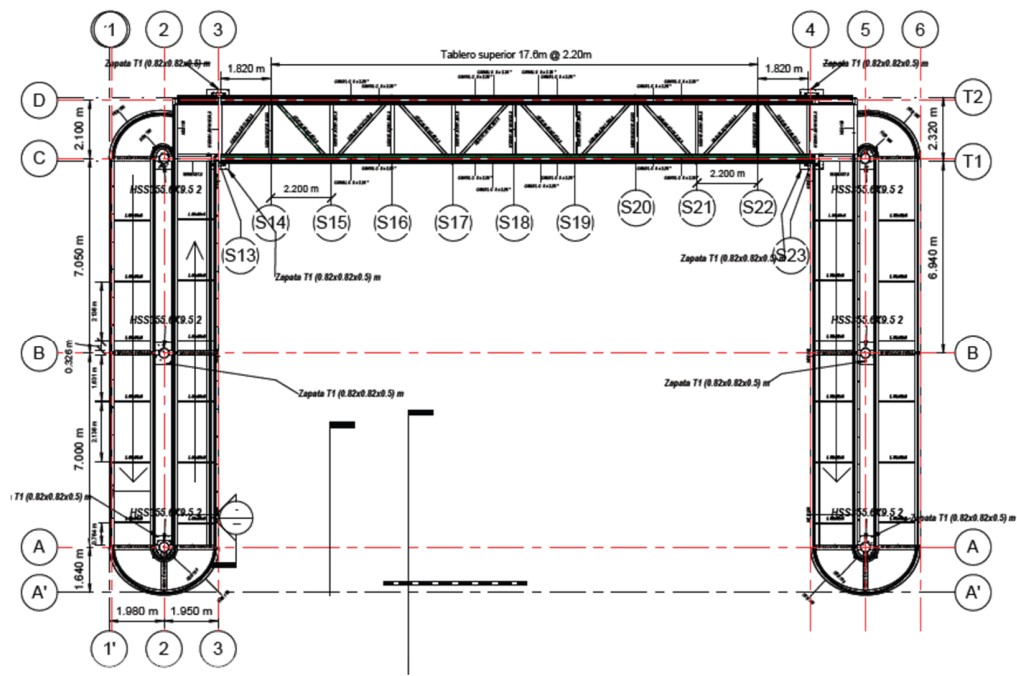

**Figure 5.** Plan view of the design of the pedestrian bridges under study (schematic view).

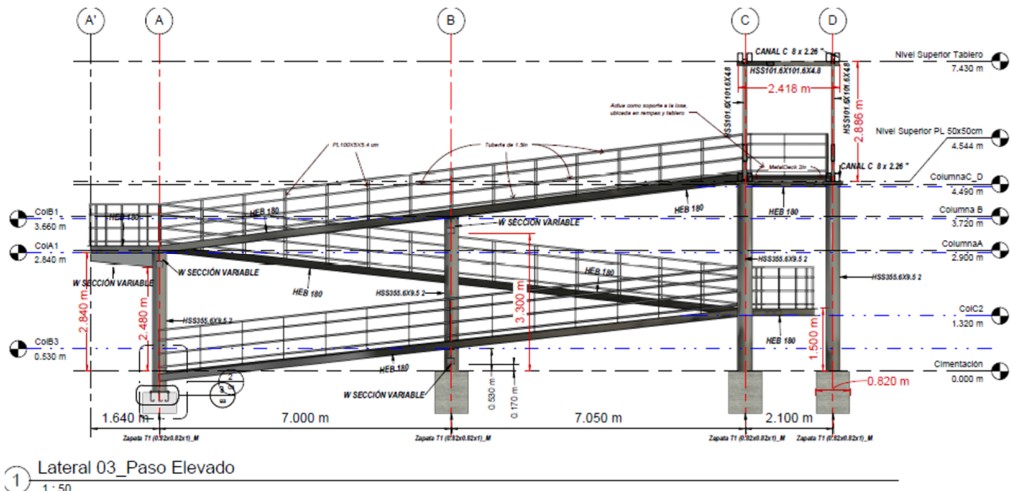

**Figure 6.** Elevation view of the design of the pedestrian bridges under study (schematic view).

**Table 1.** General description of the Uninorte, Sagrado and Unilibre bridges.

| Lateral View | Bridge Deck | Interior View | Details of View |
|---|---|---|---|
| Uninorte | | | |
| 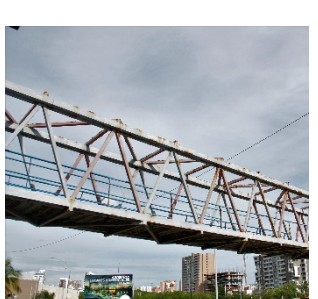 | 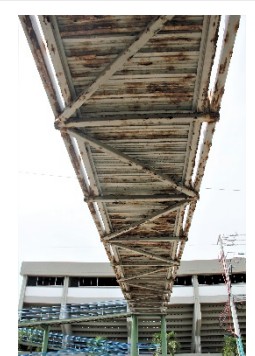 | 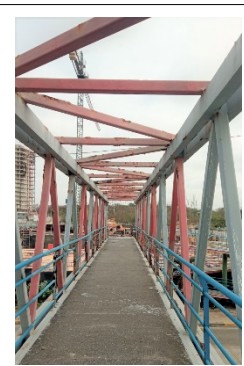 | 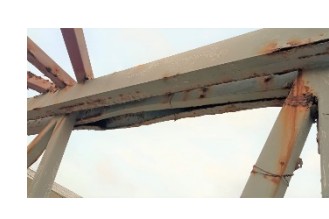 |
| Sagrado | | | |
| 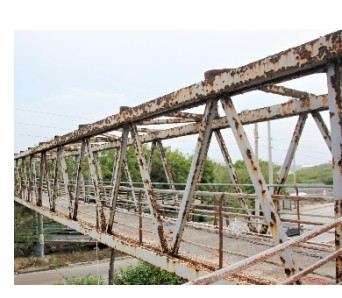 | 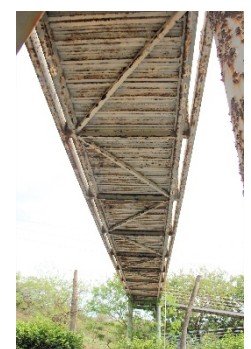 | 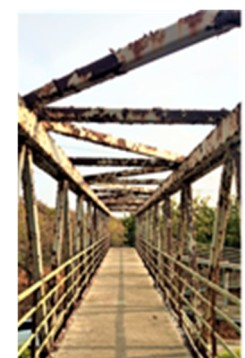 | 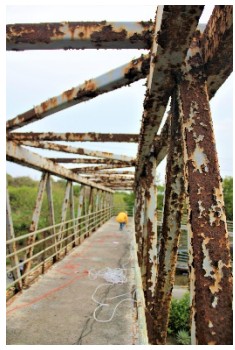 |

## 2. Literature Review

Some recent research has focused on identifying the modal parameters of structures or elements subjected to random vibrations. The effect of deterioration or degradation of some properties through changes in some dynamic parameters, such as damping, has already been explored [15,16]. Chao-Feng et al. [15] studied the influence of longitudinal reinforcement ratio, steel surface shape, stirrup spacing, and corrosion coefficient on the loss of tangent modulus of a concrete specimen-type element. They found that the reinforcement corrosion damage causes changes in stiffness, damping, and other dynamic parameters. The damping coefficient increases progressively as damage accumulates; therefore, it is feasible to use this parameter to detect and evaluate corrosion damage of concrete members. In another application, Khoshmanesh et al. [16] studied the changes in stiffness and damping properties of a thick adhesive joint for wind turbine blades during different damage phases. The stiffness was measured with an extensometer, and the

damping was calculated using vibration and thermography techniques. The conclusions reveal the potential in the monitoring changes in damping to infer incipient damage in an adhesively bonded composite joint structure.

Recent studies have proposed algorithms and equations that optimize the methods used for identification. New methods have been proposed for identifying modal parameters from the experimental data using ambient vibrations [17–19]. Bhuiyan et al. [17] presented an automated online signal reconstruction algorithm that stands out since, in practice, some sensors can induce erroneous data, causing a place of the structure that is not damaged to be identified as damaged or vice versa. In addition, Gui et al. [18] proposed three support vector machines based on optimization algorithms in the same year. Grid search, partial swarm optimization, and genetic algorithms are the techniques used to optimize the penalty parameters and Gaussian kernel function parameters. Feature extraction was conducted using time series data to capture the effective damage characteristics to propose algorithms and to monitor structural health and detect damage. In addition, Zhu & Au [19] proposed a modal identification method in the frequency domain based on Bayesian methods using asynchronous "output-only" ambient data (environmental vibration records or data that are the product of operational modal analysis (OMA)). The validation of the method is performed with synthetic and laboratory data, finding the most probable values of modal properties.

Zhu & Au [5] also published a complementary work to the one mentioned above. They investigated the uncertainty in the identification of modal parameters in terms of their posterior covariance matrix. For this purpose, analytical expressions were derived to evaluate the posterior covariance matrix accurately and efficiently. Subsequently, Zhu & Au [20] worked on defining a Bayesian model based on uncertain modal property data identified from OMA. The study establishes that the uncertainty in the identification is embedded in the posterior distribution of the modal properties of interest, considering the given data. They used a Gaussian model to describe the potentially unknown relationship between modal properties and environmental conditions and validated the method with synthetic and laboratory data. Au et al. [21] also focused on understanding the uncertainty in modal parameter identification. They proposed an "uncertainty law" for closed modes in 2021, comprising closed-form analytical expressions for the remaining uncertainty of the identified modal parameters using output-only ambient vibration data.

Another group of researchers have based their work on showing learning from structural health monitoring (SHM) experiences in different structures [22–25]. Noel et al. [22] presented a comprehensive review of damage management using wireless sensor networks. In this work, they discuss comprehensive solutions to network design problems such as scalability, time synchronization, sensor location, and data processing. In the same year, Torres et al. [23] published their research work that focused on the calibration and adjustment of finite element models of the Metropolitan Cathedral of Santiago de Chile. The model is based on identifying modal parameters from experimental data, applying methods such as enhanced frequency domain decomposition (EFDD) and stochastic subspace identification (SSI) to process and find the modal parameters of the building. In addition, Viviescas et al. [24] analyzed the behavior of the Gómez Ortiz bridge throughout its life cycle. The importance of this experience is that the bridge has a large span, thus, more notable dynamic properties. Through monitoring, changes caused by possible damage were detected to establish maintenance planning. The OMA techniques employed for data processing such as peak selection (PP), frequency domain decomposition (FDD), EFDD, and SSI were used. They found that similar data are obtained with each technique, except for damping, which confirms the uncertainty around this parameter. Likewise, Su et al. [25] reviewed the techniques and applications of SHM in super-tall structures. In this work, the techniques of vibration analysis, seismic effects monitoring, wind effects monitoring, comfort assessment, temperature effects monitoring, and construction monitoring of supertall structures were described and summarized.

Finally, when considering the advantages of OMA and SHM applications, it is evident that these methodologies allow faster identification of damage in a structure in a non-

invasive way [23]. For this reason, recent research works have focused on implementing these methodologies at a lower cost [3,4]. Duarte & Ortiz [3] quantified the uncertainties of the modal parameters of a flexible structure instrumented with traditional measurement devices (piezoelectric sensors) and with low-cost sensors (smartphones and Raspberry Pi microprocessors). The applied method used a Bayesian approach and modal update techniques to determine the probability density functions (PDF) of the modal parameters. Under this same low-cost approach, Caballero-Russi et al. [4] presented a low-cost SHM system's design, implementation, and validation with experimental tests in the laboratory. The system comprises a wireless sensor network (WSN) and a base station. System WSN validation is performed with a traditional wired SHM system at a small scale. The vibration data recorded with both systems were processed for the identification of dynamic parameters such as natural frequency, vibration modes, and damping ratios [4].

Other investigations related to the topics addressed in this work focus on methods and techniques for identifying corrosion. The authors propose a practical, fast, and economical method that does not require complex calculations or large equipment and allows the structures to be characterized. Table 2 shows the advantages and disadvantages of some commonly used methods to identify corrosion versus the proposed method.

**Table 2.** Current methods for corrosion degree assessment.

| Technique or Method | Advantages | Disadvantages |
|---|---|---|
| Immersion corrosion testing of metals [26] | * Online monitoring technique.<br>* Direct measurement of metal loss.<br>* Allows to calculate the overall corrosion rate. | * Rapidly promoting the corrosion of the elements does not represent the real state of the structures. The controlled experiment in the laboratory simulates an approximate behavior of the phenomenon; however, there may be variables that are not adequately simulated, creating an environment different from the real one.<br>* Expensive.<br>* Long experimentation periods.<br>* The behavior of certain metals and alloys can be influenced by dissolved oxygen.<br>* The accumulation of corrosion products can influence the composition of the fluid that participates as a corrosive agent. |
| Thermography [27,28] | * Allows to observe the distribution of corrosion in the element under study.<br>* Non-Destructive techniques (NDTs)<br>* Post-processing of temperature maps can be done using AI.<br>* Growing trends in obtaining quantitative or qualitative data through AI (image processing). | * Thermography cannot detect the corrosion rate.<br>* Quantitative image analysis.<br>* Expensive.<br>* High computational cost of image post-processing.<br>* It does not quantify the degree of corrosion; it only thermally characterizes the anomalies detected.<br>* The image quality is affected by the high price since the resolution is limited to the available budget.<br>* The camera's field of view may not apply to large infrastructures.<br>* High dependence on weather conditions.<br>* High dependence on the experience of the camera operator. |
| Acoustic emission (AE) [28,29] | * Real-time control method in laboratory experiments.<br>* Non-destructive techniques (NDTs).<br>* Prediction methods. | * For real structures monitoring, there is some difficulty in "in situ" real structure corrosion damage monitoring using the AE technique.<br>* Rapidly promoting the corrosion of the elements does not represent the real state of the structures. The controlled experiment in the laboratory simulates an approximate behavior of the phenomenon; however, there may be variables that are not adequately simulated, creating an environment different from the real one. |
| Visual auscultation + corrosion matrix [this research] | * Low-cost<br>* Rapid<br>* Easy implementation<br>* Almost immediate results | * In the calculation of corroded surfaces per element, approximations are made.<br>* It depends on the observer.<br>* Measurements of the reduction of thicknesses due to corrosion are not performed.<br>* Quantitative and global values of the degree of corrosion of the structure are obtained from a qualitative assessment of the consequence of the observed deterioration and the value of the total corroded surface expressed in [%].<br>* The degree of corrosion is expressed as a range due to the uncertainty of measurements that are not detailed and specific. |

### 3. Methodology

In this research, three lines of work were identified to provide the necessary information to obtain the dynamic parameters of bridges and perform a comparative analysis. The first line corresponds to the bridge's geometry, i.e., the structural survey is performed for the numerical model construction. The second line corresponds to the determination of the theoretical dynamic parameters of each bridge obtained from a computational model; thus, each model is compared with the experimental data. Since the core of this research is the characterization of the dynamic parameters, the processing of the experimental vibration records of each bridge is performed and obtained with a portable data acquisition system (dynamic identification, Figure 7).

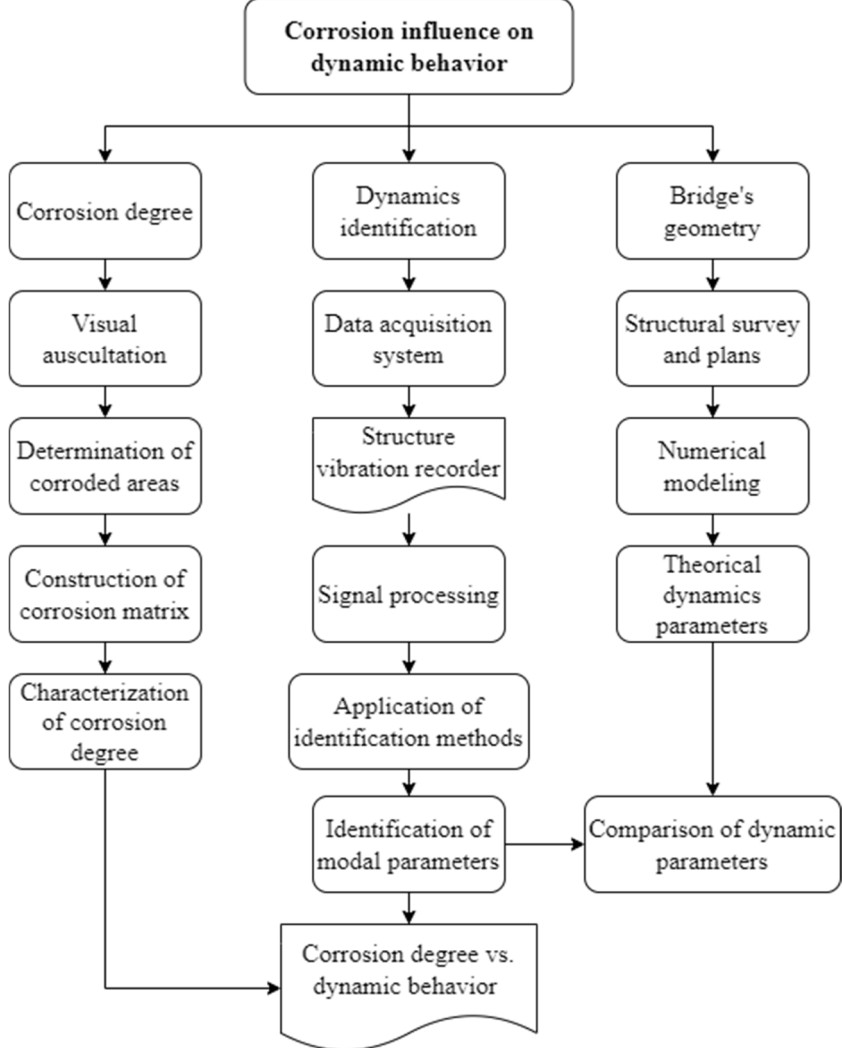

**Figure 7.** Flow chart—Research methodology.

The third line is the characterization of the corrosion degree, which is obtained through the corrosion matrix built for the authors (Section 3.3). With the knowledge of the percentage of corroded areas and the most probable consequence of the damage, it is possible to assess the generalized degree of corrosion in the structure by selecting a color in the matrix, thus determining a range of values for this variable.

### 3.1. Geometry of the Bridges

The geometric description of the structures under study is based on structural surveys on each bridge. Measurements of the section, the thicknesses of each element, and the lengths are made. The connections and support elements are described, and the structural

plans are built with all the information collected. Finally, three numerical models are built, one for each bridge. From the modeling, the dynamic parameters of each structure are obtained, such as the vertical and lateral frequency for the first four vibration modes. With these results, it is possible to measure the order of magnitude of the frequency values in each of the bridges to compare.

### 3.2. Dynamic Characterization of Structures

3.2.1. Description of the Test

The measurement of dynamic parameters was performed using uniaxial and triaxial piezoelectric sensors (PCB Piezotronics). These devices have a sensitivity of 1000 mV/g, a measurement range of $\pm 5.0$ V, and a frequency range of $\pm 5.0\%$ from 0.5–3000 Hz. The acquisition system comprises an NI CompactDAQ data platform (National Instruments) with NI 9234 acquisition cards (sample rate of 1650 Hz). Coaxial cables capture and transfer the acceleration input (Figure 8) [4].

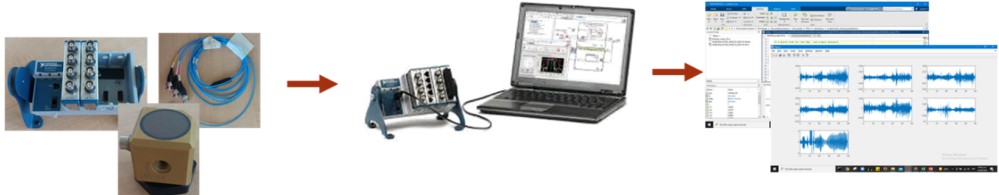

**Figure 8.** Measurement and data acquisition system.

Five nodes were identified on each structure under study at distances proportional to the length of the deck. Measurements were taken in the gravitational (Z) and transverse (X) directions of the deck (Figure 9).

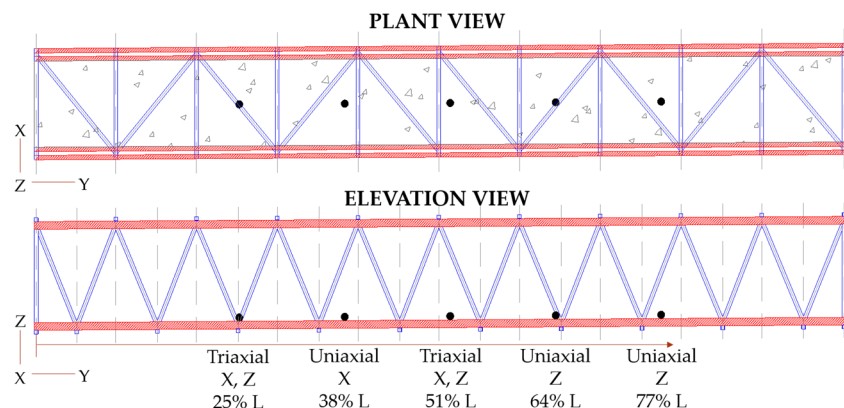

**Figure 9.** Experimental setup (location of accelerometers).

3.2.2. Signals Processing and Experimental Data

According to the problem statement and the proposed methodology, the objective of this study is to obtain time series from each experimental unit (Uninorte, Sagrado and Unilibre). The sampling frequency of the acquisition equipment allows to have a large number of samples in a few minutes (99,000 data in one minute), which provides a good spectral resolution if measurement times of 5 to 10 min are considered.

On the other hand, according to the reference values of the frequencies of each bridge, it is necessary to resample the signal, obtain a better spectral resolution, reduce the bandwidth, and identify frequencies lower than 10 Hz. The Python signal.resample() function from the SciPy library was applied to each signal to be processed. Subsequently, the experimental power spectral density (PSD) was calculated, which shows the resonant frequencies.

### 3.2.3. Identification of Dynamic Characteristics of the Bridges

The dynamic properties of each bridge were identified by fitting the analytical PSD to the experimental PSD using optimization. The implemented method minimized the error by least squares. An objective function (Equation (1)) is used to obtain the most optimal value of the parameters, given the experimental data. The above equation is the analytical PSD function of a single degree-of-freedom system.

$$\hat{S}_k = SD_k + S_e \tag{1}$$

$$D_k = \left[\left(\beta_k^2 - 1\right)^2 + (2\zeta\beta_k)^2\right]^{-1}; \beta_k = f/f_k \tag{2}$$

where $f$ is the natural frequency, $\zeta$ is the damping ratio, $S$ is the spectral density of the vibration mode, and $Se$ is the PSD of the noise [3].

### 3.3. Characterization of the Degree of Corrosion

According to the recommendations of the Corrosion College, the corrosion of a structure can be evaluated in an easy and practical way, relating the severity of the exposure environment to the consequences of deterioration [30]. As a result, a matrix with a color scale indicating the degree of corrosion of the structure is obtained. The authors took the Corrosion College corrosion matrix as a basis, considering only one exposure environment according to Table F.5.2.4-2 of NSR-10 [31].

The corrosion matrix (Figure 10) is built to obtain a quantitative and generalized value of the degree of corrosion. Thus, the corroded area of the structure is related in percentage [%] with the consequences of the deterioration observed, considering the exposure of the three bridges to a non-industrial marine environment.

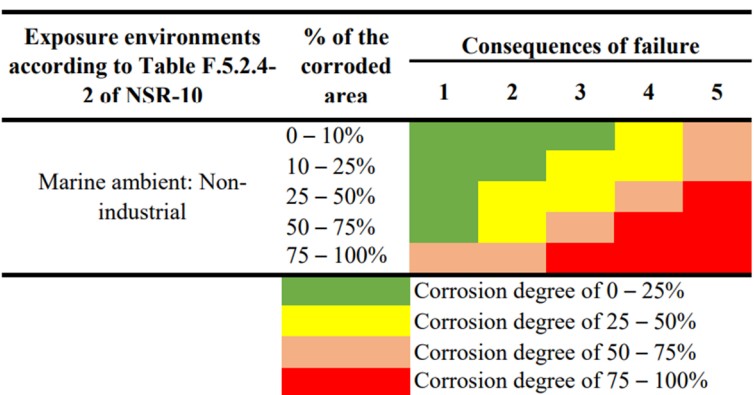

**Figure 10.** Corrosion matrix.

The corrosion matrix is used through the approximate assessment of corroded areas, element by element, for each bridge under study. Then, a weighted average is established for all structural elements, thus determining a percentage of corroded area characteristic of the bridge. A generalized degree of corrosion of each structure is obtained by relating the percentage of the corroded area and the possible consequences of failure (Section 4.1).

The consequences of failure in terms of corrosion deterioration, which have been considered, are as follows:

1. It does not affect the functionality and structural integrity.
2. Higher maintenance costs will be incurred.
3. It partially affects functionality due to local damage.
4. It completely affects functionality and puts structural integrity at risk.
5. Structural integrity is at risk.

## 4. Results

The first bridge measurements were carried out in March 2022, starting with the structural survey. This was followed in parallel by vibration measurements, visual auscultation, and photographic recording to calculate the corrosion degree.

Experimental tests of environmental vibrations were carried out for the dynamic characterization, obtaining 1,981,936 samples for each bridge, according to the proposed methodology. The results obtained are presented below.

### 4.1. Bridge Geometry and Numerical Modeling

#### 4.1.1. Bridges Geometry Results

A structural survey was performed to define the geometry of the bridges. With this information, drawings and numerical modeling were completed. It was identified that the geometry of the bridges is similar since the cross-sections of the elements for the upper and lower chords are the same (steel built-up box beam composed of two C-sections welded face-to-face). In addition, the secondary elements (diagonals and struts) in the vertical and horizontal plans (all detailed with a $103 \times 103 \times 4$ mm structural steel tube) are the same. However, the difference consists in the lower struts of the Unilibre bridge detailed with a $58 \times 200 \times 4$ mm C-section. Nonetheless, the general geometry of the structures under study is similar. Figure 11 shows a schematic view of the lateral, transverse and plan views of the bridges for identifying each element, and Table 3 shows the results of the measurements of elements on each bridge.

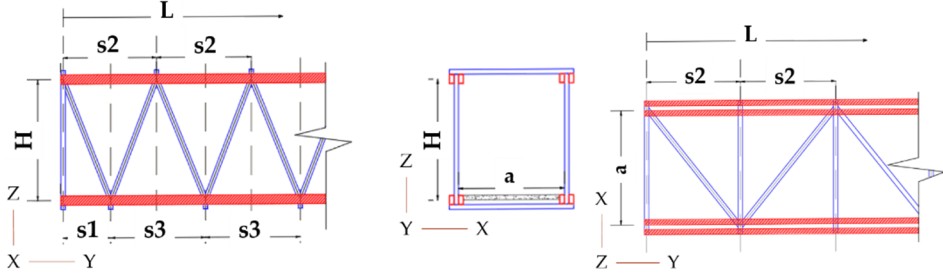

**Figure 11.** Geometry and cross-section of bridges (lateral, transverse and plan views).

**Table 3.** Bridges geometry.

| Bridges | Geometry [m] | | | | | |
|---|---|---|---|---|---|---|
| | a | L | H | s1 | s2 | s3 |
| Uninorte | 2.50 | 22.47 | 2.66 | 1.12 | 2.20 | 2.24 |
| Sagrado | 2.00 | 21.93 | 2.66 | 0.80 | 1.85 | 2.25 |
| Unilibre | 2.01 | 26.13 | 2.36 | 0.38 | 2.32 | 2.22 |

#### 4.1.2. Numerical Modeling Results

The numerical models are implemented in ETABS 2015 Ultimate V 15.2.2. The models are based on the steel bridge's geometry, the transverse section of its elements and the supporting condition. The numerical models include the deck with its elements and a simple support condition.

First, an initial model was constructed to obtain reference values of the modal parameters. Later, these parameters were updated with the experimental modal values for each bridge. This adjustment is only implemented for the first mode because the principal frequency of the bridges was specifically studied.

Figure 12 shows the results of the analytical models. Each column depicts the first four vertical vibration modes of each bridge. Table 4 shows the frequency values associated with the first four vertical vibration modes, in the three bridges modeled for the initial numerical model.

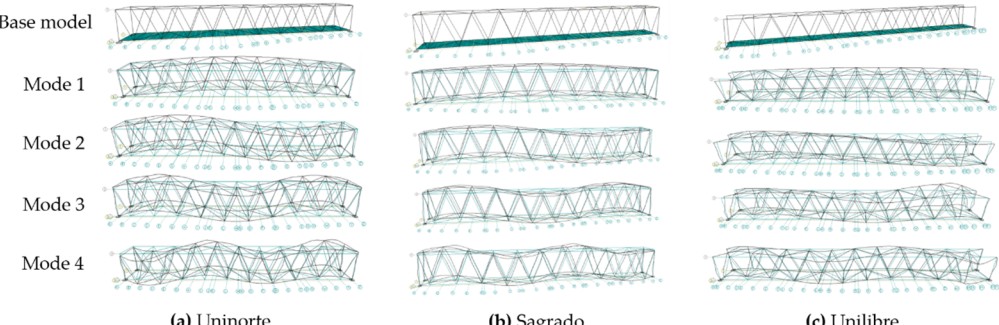

**Figure 12.** First four vertical modes of vibration obtained from the initial numerical model of the bridges.

**Table 4.** Frequencies associated with the first four vertical vibration modes in [Hz] of the initial analytical model.

| Bridge | Frequencies [Hz] | | | |
|---|---|---|---|---|
| | **Mode 1** | **Mode 2** | **Mode 3** | **Mode 4** |
| Uninorte | 6.32 | 20.41 | 30.73 | 39.38 |
| Sagrado | 7.35 | 23.50 | 34.60 | 43.71 |
| Unilibre | 7.14 | 21.60 | 31.06 | 38.79 |

The computational time for each analytical model is lower to one second (1 s). The hardware comprised a 12th Gen Intel(R) Core (TM) i7-1255U 1.70 GHz processor, with 16 GB of memory RAM under the Windows 10 Pro operative system.

### 4.2. System Identification Results

The identification system corresponds to each bridge analyzed. Its dynamic characterization is calculated by optimizing Equation (1) fitted to the experimental PSD. The method used for the fitting is the mean square error, minimizing the function. Figure 13 shows the results of the signals processing, specifically the power spectral density for each bridge, which is obtained with the bridge records in the vertical and transverse directions (Z and X directions, respectively—Figure 11).

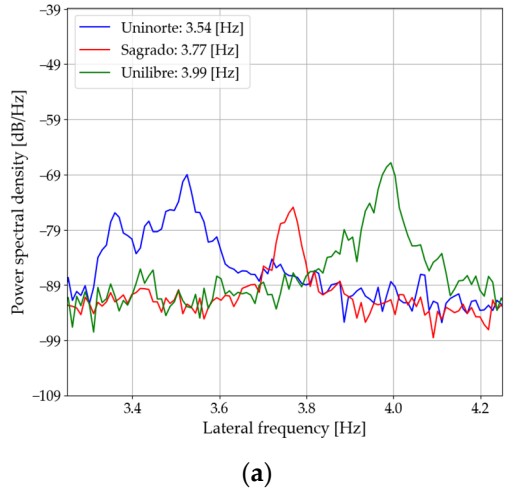

(**a**)

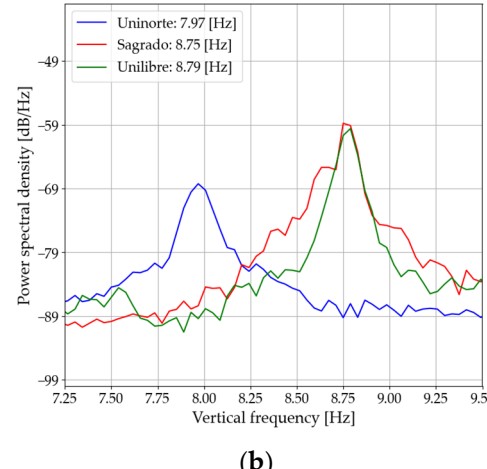

(**b**)

**Figure 13.** Power spectral density of experimental frequencies of the bridges: (**a**) lateral and (**b**) vertical.

The signal processing includes clean signs and determining logs to work. To dynamically characterize a structure, an adequate spectral resolution is necessary. However, if a short record of seconds is obtained, it is possible to characterize the structure (Figures 13–19).

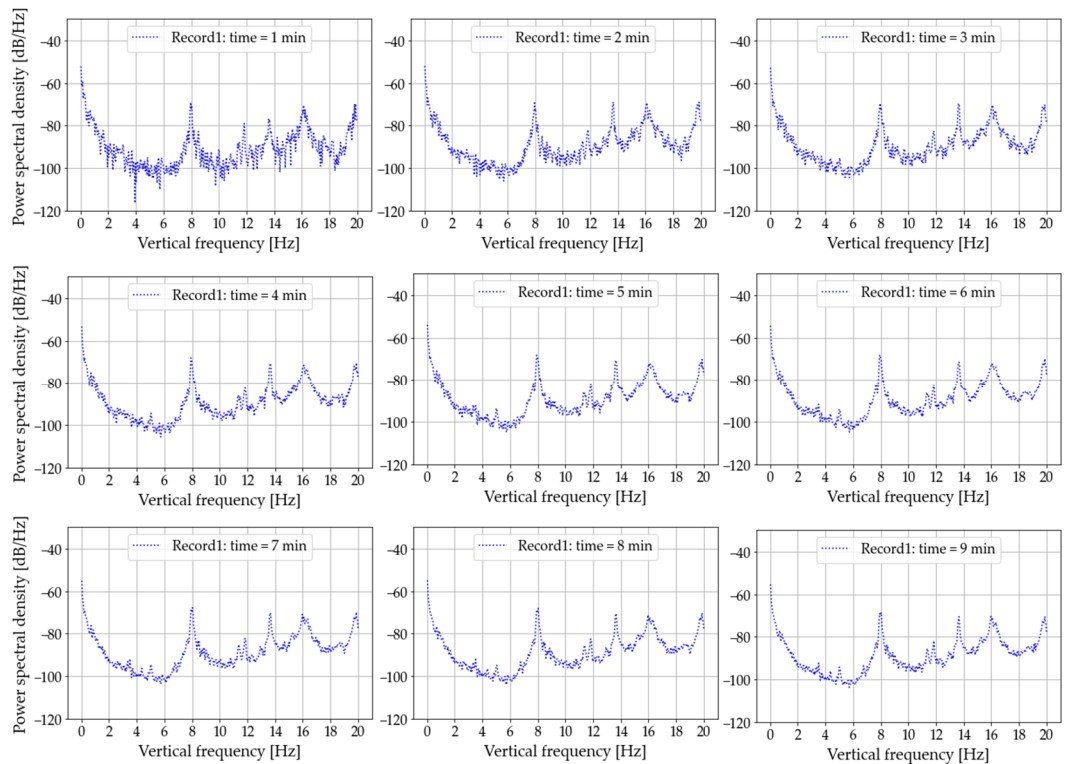

**Figure 14.** Power spectral density of the first nine vertical frequency observations of the Uninorte bridge.

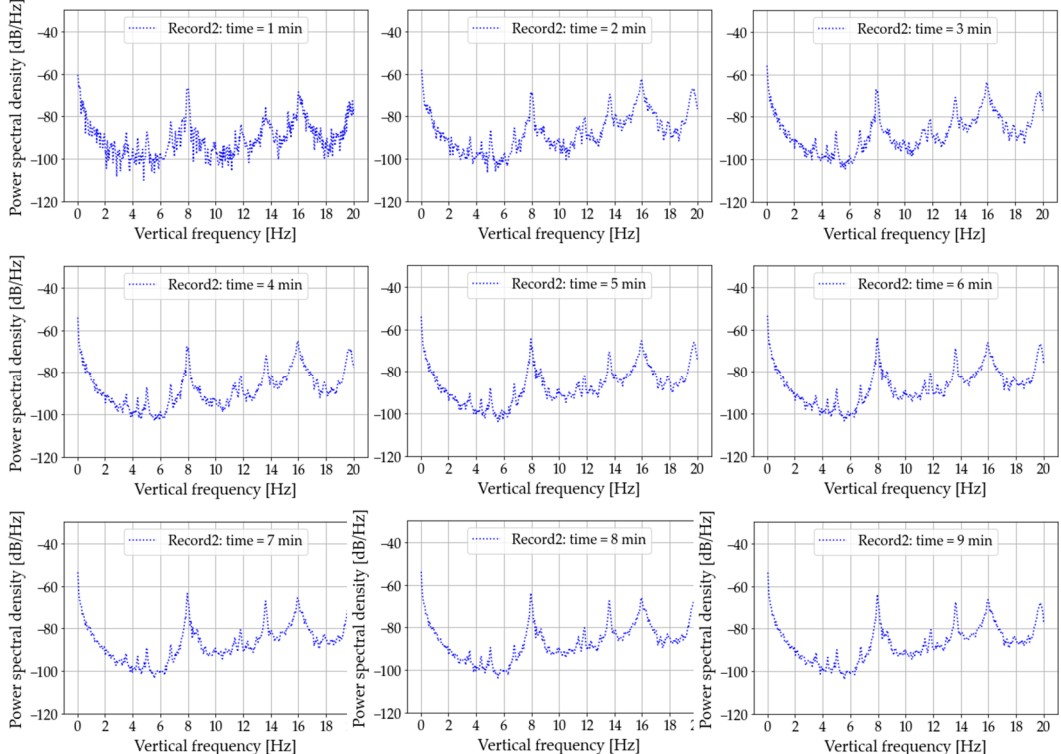

**Figure 15.** Power spectral density of the last nine vertical frequency observations of the Uninorte bridge.

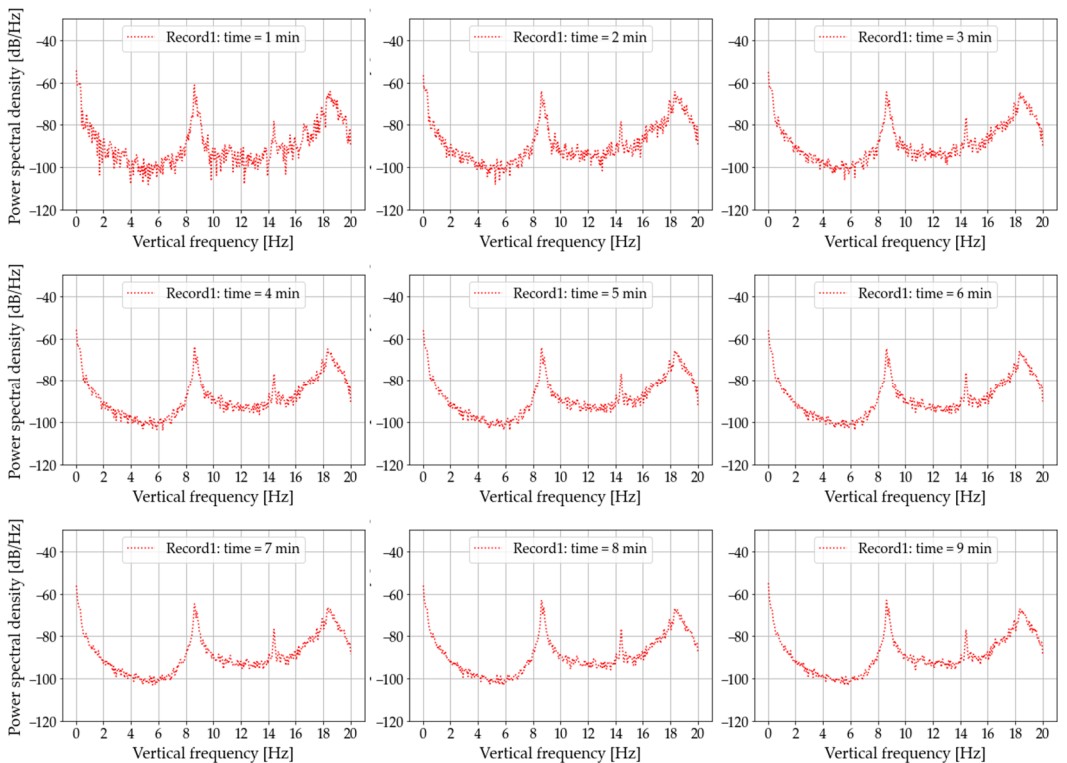

**Figure 16.** Power spectral density of the first nine vertical frequency observations of the Sagrado bridge.

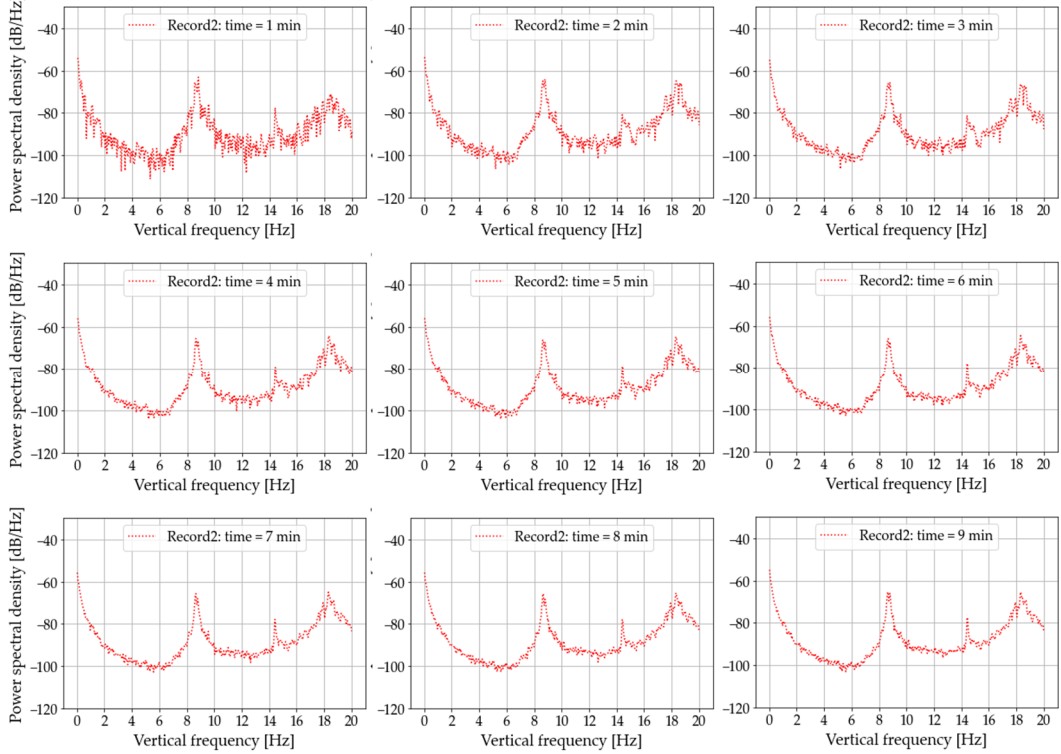

**Figure 17.** Power spectral density of the last nine vertical frequency observations of the Sagrado bridge.

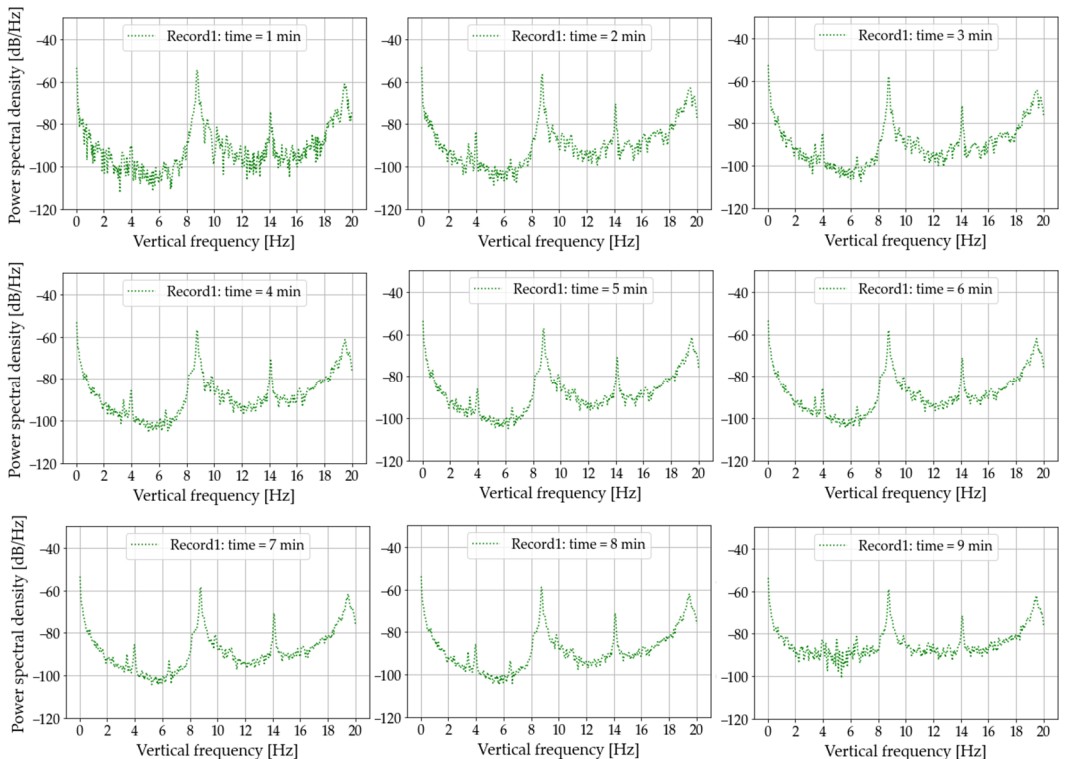

**Figure 18.** Power spectral density of the first nine vertical frequency observations of the Unilibre bridge.

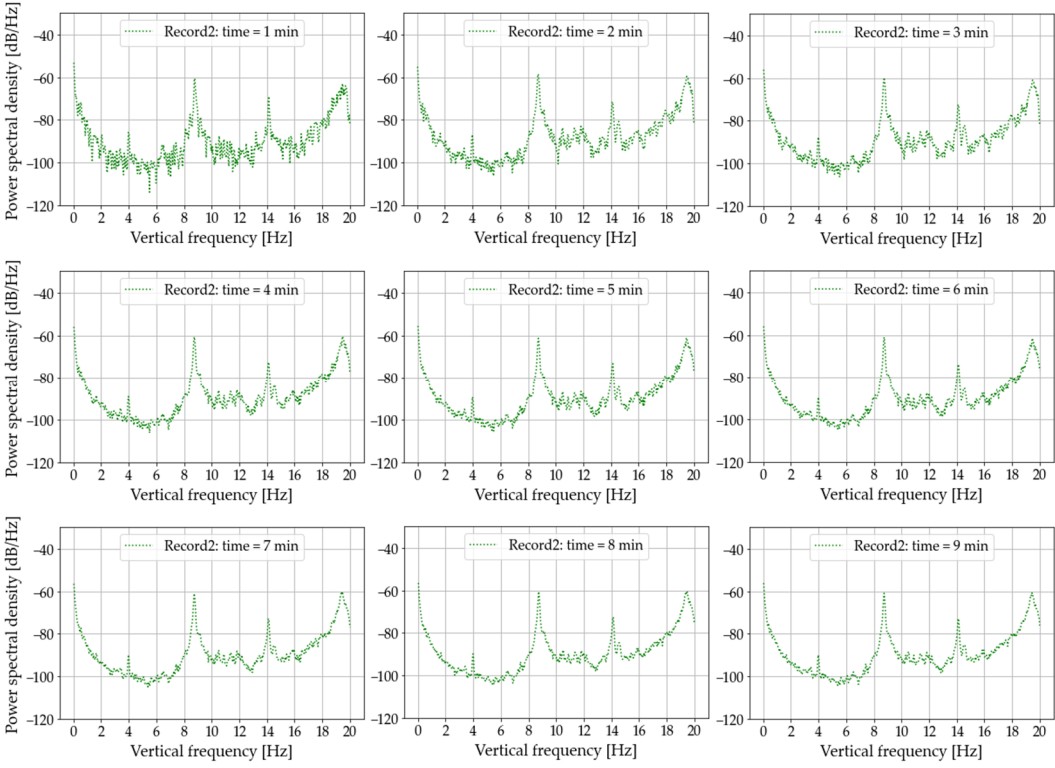

**Figure 19.** Power spectral density of the last nine vertical frequency observations of the Unilibre bridge.

*4.3. PSD Curve Fit Results*

The modal parameters shown in Table 5 for each bridge are obtained using curve fitting optimization algorithms (Figure 20), as shown in Section 3.2.3. Equation (1) is optimized according to the experimental data, obtaining the optimal values for the dynamic

parameters such as natural frequency, damping, spectral density of vibration mode and spectral density of noise.

**Table 5.** Results of the curve fit for each bridge.

| Bridge (Experimental Unit) | Observation | f [Hz] | ζ [%] | log(Se) [dB/Hz] | log(S) [dB/Hz] |
|---|---|---|---|---|---|
| Uninorte | 1 | 7.98 | 0.12 | −21.03 | −26.48 |
| | 2 | 7.97 | 0.28 | −20.60 | −25.96 |
| | 3 | 7.97 | 0.32 | −20.65 | −25.70 |
| | 4 | 7.96 | 0.29 | −20.59 | −25.56 |
| | 5 | 7.98 | 0.29 | −20.58 | −25.41 |
| | 6 | 7.98 | 0.28 | −20.60 | −25.49 |
| | 7 | 7.99 | 0.35 | −20.47 | −25.05 |
| | 8 | 7.99 | 0.34 | −20.51 | −25.15 |
| | 9 | 7.99 | 0.35 | −20.52 | −25.17 |
| | 10 | 7.99 | 0.15 | −20.99 | −25.45 |
| | 11 | 7.99 | 0.36 | −20.17 | −25.14 |
| | 12 | 7.99 | 0.30 | −20.24 | −25.08 |
| | 13 | 7.99 | 0.33 | −20.33 | −25.18 |
| | 14 | 7.99 | 0.43 | −20.10 | −24.28 |
| | 15 | 7.99 | 0.36 | −20.09 | −24.33 |
| | 16 | 7.98 | 0.32 | −20.05 | −24.35 |
| | 17 | 7.98 | 0.33 | −20.09 | −24.40 |
| | 18 | 7.98 | 0.32 | −20.10 | −24.48 |
| Sagrado | 1 | 8.64 | 0.21 | −20.79 | −24.56 |
| | 2 | 8.66 | 0.37 | −20.62 | −24.45 |
| | 3 | 8.68 | 0.39 | −20.50 | −24.41 |
| | 4 | 8.68 | 0.43 | −20.50 | −24.19 |
| | 5 | 8.68 | 0.47 | −20.56 | −24.15 |
| | 6 | 8.68 | 0.46 | −20.61 | −24.23 |
| | 7 | 8.69 | 0.47 | −20.65 | −24.11 |
| | 8 | 8.69 | 0.44 | −20.69 | −24.03 |
| | 9 | 8.69 | 0.44 | −20.65 | −24.09 |
| | 10 | 8.72 | 0.65 | −21.01 | −23.66 |
| | 11 | 8.71 | 0.51 | −20.62 | −23.76 |
| | 12 | 8.71 | 0.48 | −20.74 | −24.12 |
| | 13 | 8.70 | 0.47 | −20.60 | −24.22 |
| | 14 | 8.70 | 0.47 | −20.64 | −24.32 |
| | 15 | 8.70 | 0.43 | −20.62 | −24.43 |
| | 16 | 8.69 | 0.42 | −20.56 | −24.45 |
| | 17 | 8.69 | 0.47 | −20.54 | −24.32 |
| | 18 | 8.71 | 0.47 | −20.53 | −24.19 |
| Unilibre | 1 | 8.77 | 0.00 | −21.62 | −23.55 |
| | 2 | 8.77 | 0.00 | −20.91 | −23.82 |
| | 3 | 8.77 | 0.00 | −20.92 | −24.22 |
| | 4 | 8.78 | 0.23 | −20.67 | −23.12 |
| | 5 | 8.78 | 0.25 | −20.67 | −23.19 |
| | 6 | 8.78 | 0.25 | −20.68 | −23.39 |
| | 7 | 8.78 | 0.23 | −20.69 | −23.55 |
| | 8 | 8.77 | 0.22 | −20.72 | −23.66 |
| | 9 | 8.77 | 0.21 | −19.63 | −23.72 |
| | 10 | 8.78 | 0.11 | −20.98 | −24.81 |
| | 11 | 8.76 | 0.17 | −20.53 | −23.89 |
| | 12 | 8.76 | 0.16 | −20.71 | −24.20 |
| | 13 | 8.76 | 0.21 | −20.72 | −24.22 |
| | 14 | 8.76 | 0.19 | −20.80 | −24.43 |
| | 15 | 8.76 | 0.16 | −20.87 | −24.58 |
| | 16 | 8.75 | 0.22 | −20.82 | −24.51 |
| | 17 | 8.75 | 0.22 | −20.80 | −24.44 |
| | 18 | 8.75 | 0.21 | −20.83 | −24.54 |

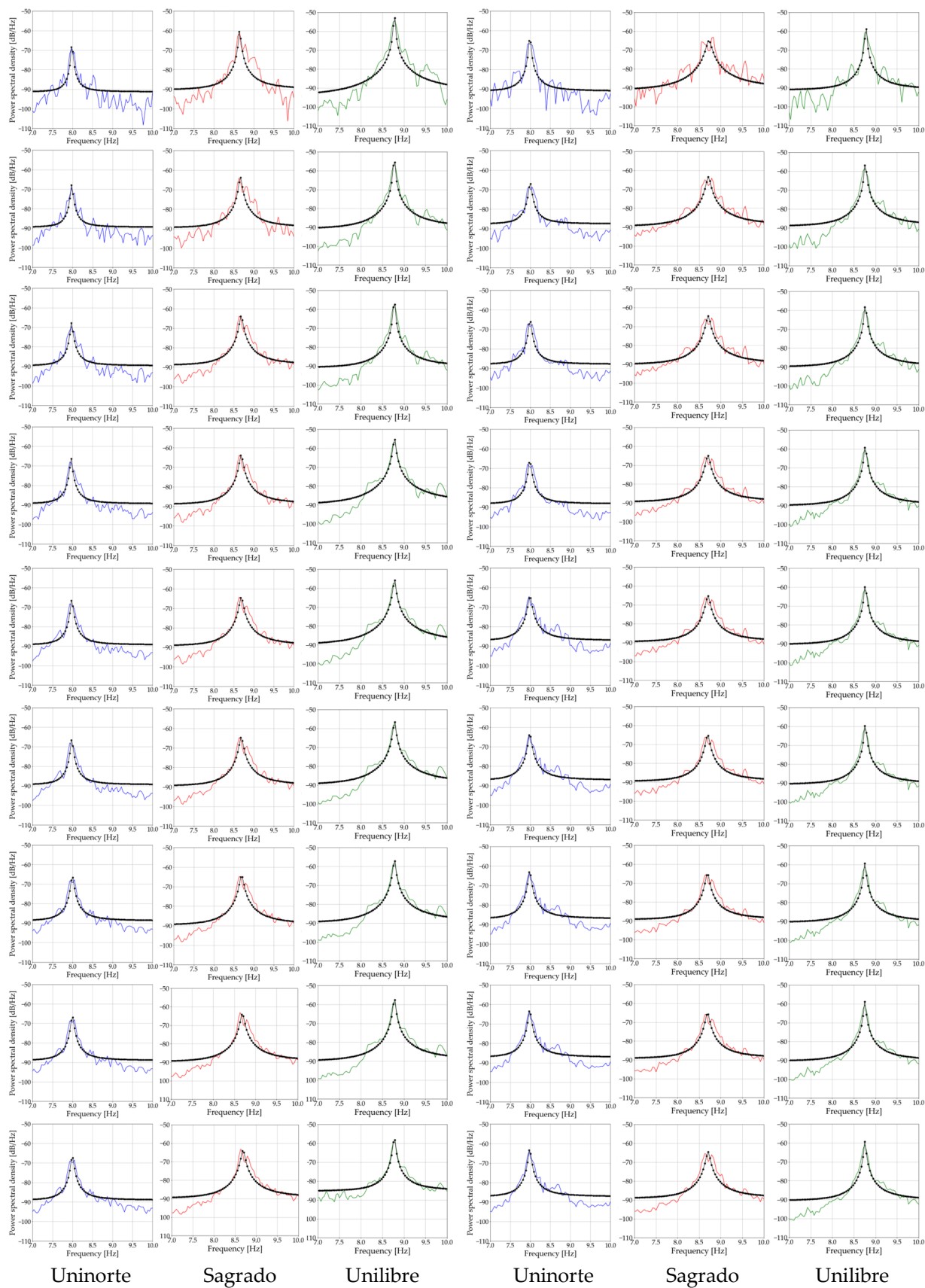

**Figure 20.** Curve fitting of experimental PSD for the observations in each bridge.

From the results obtained in Table 5, a mean value of the dynamic parameters is obtained and recorded in Table 6. The vertical frequency corresponding to the first mode of vibration is used to adjust the numerical model. The theoretical modal values are sought to be similar to the experimental modal values.

**Table 6.** Average curve fitting results for each bridge.

| Bridge (Experimental Unit) | f [Hz] | ζ [%] | log(Se) [dB/Hz] | log(S) [dB/Hz] |
|---|---|---|---|---|
| Uninorte | 7.984 | 0.307 | −20.428 | −25.148 |
| Sagrado | 8.690 | 0.446 | −20.634 | −24.205 |
| Unilibre | 8.769 | 0.167 | −20.749 | −23.959 |

Updating Numerical Model Based on Fitted PSD

Table 7 shows the fitted frequency values for the first four vertical vibration modes. These values correspond to the numerical model update. The adjustment is made based on the experimental frequency of the first mode.

**Table 7.** Fitted frequencies associated with the first four vertical vibration modes in [Hz].

| Bridge | Fitted Frequencies [Hz] | | | | Adjustment Factor | Error [%] |
|---|---|---|---|---|---|---|
| | Mode 1 | Mode 2 | Mode 3 | Mode 4 | | |
| Uninorte | 8.02 | 26.94 | 40.18 | 44.78 | 1.90 | 0.48 |
| Sagrado | 8.73 | 27.62 | 40.84 | 49.89 | 1.50 | 0.41 |
| Unilibre | 8.75 | 25.06 | 35.98 | 43.47 | 1.80 | 0.21 |

*4.4. Degree of Corrosion*

Table 8 shows the results of the visual auscultation of the bridges. The corroded areas were determined by calculating the deteriorated area with respect to the total area. This process is performed for each element and in each bridge. Finally, a weighted average of the corroded areas of a structure is obtained in percentage.

The percentage of corroded area and the consequence of deterioration are the variables, which are needed to use the corrosion matrix (Figure 21). Figure 22 shows the corrosion degree in percentage for each bridge. It was obtained using the corrosion matrix according to the proposed methodology.

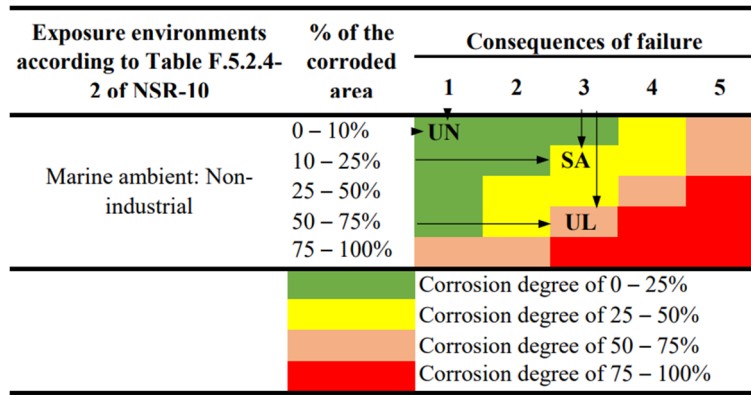

**Figure 21.** Evaluation of the corrosion degree. With UN: Uninorte, SA: Sagrado and UL: Unilibre.

**Figure 22.** Characterization of bridge corrosion.

**Table 8.** Fitted frequencies associated with the first four vertical vibration modes in [Hz].

| Element Type | Uninorte Bridge | | | Sagrado Bridge | | | Unilibre Bridge | | |
|---|---|---|---|---|---|---|---|---|---|
| | Total Surface [m²] | Corroded Area [%] | Total Corroded Surface [m²]/ Element | Total Surface | % Corroded Area | Total Corroded Surface/ Element | Total Surface | % Corroded Area | Total Corroded Surface/ Element |
| Upper chords | 57.16 | 8 | 4.57 | 55.79 | 18 | 10.04 | 66.47 | 56 | 37.23 |
| Lower chords | 57.16 | 10 | 5.72 | 55.79 | 24 | 13.39 | 66.47 | 65 | 43.21 |
| Lateral diagonals | 50.40 | 5 | 2.52 | 50.40 | 30 | 15.12 | 42.68 | 60 | 25.61 |
| Upper diagonals | 12.76 | 8 | 1.02 | 12.76 | 20 | 2.55 | 12.29 | 55 | 6.76 |
| Lower diagonals | 11.49 | 7 | 0.75 | 11.49 | 20 | 2.30 | 11.06 | 60 | 6.64 |
| Upper struts | 10.97 | 4 | 0.44 | 10.97 | 25 | 2.74 | 11.02 | 50 | 5.51 |
| Lower struts | 11.97 | 5 | 0.54 | 11.97 | 25 | 2.99 | 14.87 | 50 | 7.43 |
| Vertical struts | 4.02 | 1 | 0.04 | 4.02 | 22 | 0.88 | 5.32 | 55 | 2.93 |
| Total surface bridge: | 215.93 | — | 15.59 | 213.19 | — | 50.02 | 230.18 | — | 135.30 |
| % of the corroded area/bridge: | 15.59/215.93 = 7.22% | | | 50.02/213.19 = 23.46% | | | 135.30/230.18 = 58.78% | | |

*4.5. Relationship between the Dynamic Properties of the Bridges and the Degree of Corrosion Detected*

Figure 23 shows the bivariate pairwise distributions of the dataset of the three bridges. The dispersion matrix shows the pairwise relationship between the different variables. Correlation plots were made using the Python seaborn library for data visualization.

The obtained results demonstrate that the damping ratio varies in the presence of damage in the structures, as evidenced in the literature [15,16], while the natural frequency does not vary significantly (Figure 23).

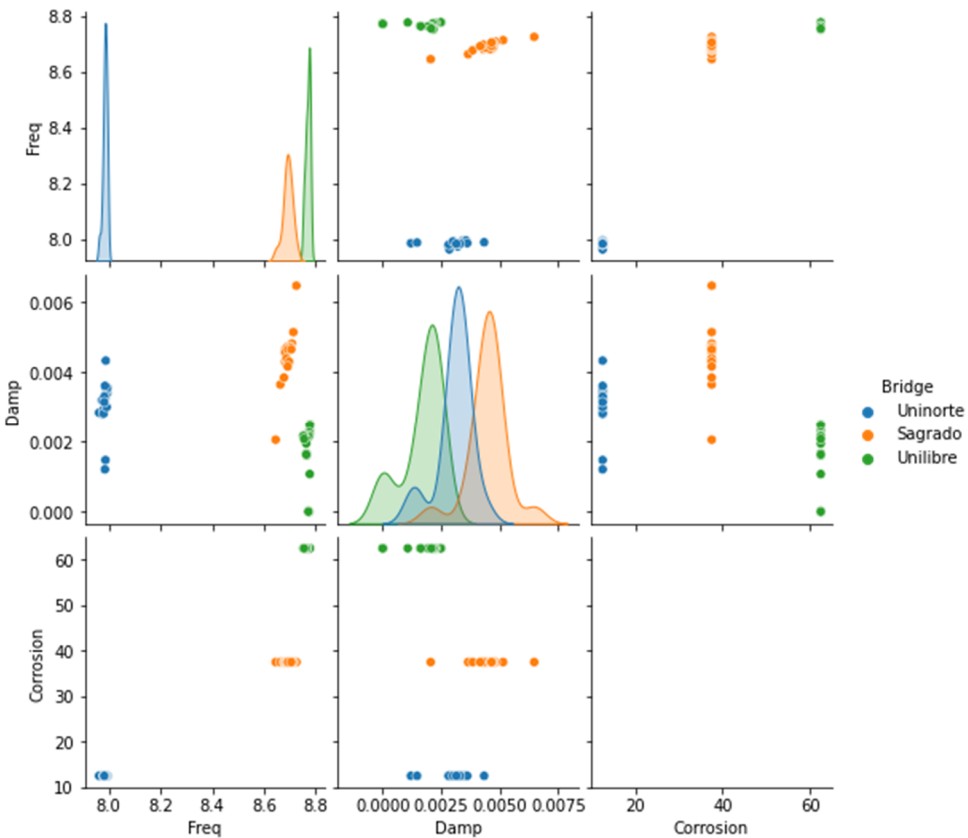

**Figure 23.** Relationship between the dynamic parameters and corrosion grade.

## 5. Discussion and Conclusions

This study evaluated the relationship between the degree of corrosion and the dynamic properties of three full-scale steel bridges. The objective of this study was to estimate if the proposed methodology for identifying the dynamic parameters is appropriate to evaluate the corrosion state of a structure. A corrosion assessment of three bridges was performed. It was found that the Unilibre bridge is the most corroded, the Sagrado bridge has a regular condition, and the Uninorte bridge is in the best state of conservation.

The results of geometry suggest that Uninorte and Sagrado are very similar (Table 3), and Unilibre is the one that presents the greatest differences in terms of geometry and mass. The Unilibre bridge shows changes in the length between supports, which is 16.29% greater than the others; however, its theoretical and experimental frequencies are close to those of Sagrado (Tables 6 and 7). The damping ratio of Unilibre differs by 62.56% with respect to the Sagrado bridge.

Additionally, it should be considered that the frequency value for the Uninorte bridge reported in a study in 2019 [3] was 8.736 Hz, which means a reduction of 8.61% if compared with the frequency identified in the present study (Table 6). This indicates that the Uninorte bridge has changed its dynamic properties after approximately three years, showing changes in its natural period from 0.114 to 0.126 s, which may indicate a loss of stiffness.

Higher values of damping ratios were observed in bridges with better maintenance against corrosion. There was no correlation between the natural frequencies of the bridge and the corrosion level measured. This shows that the damping could be a performance indicator of the state of the damage induced by corrosion, even more than the natural frequencies of the structure. The experimental damping ratio represents a more sensitive measure of damage than the natural frequency when corrosion damage is already evident in the structure. However, this parameter shows a greater dispersion than the natural frequency, which has a small variability from one sample to another (Figure 23).

Finally, considering the results of the Uninorte bridge and the study performed in 2019, and according to the approach of recent research [32,33], it is recommended that future works should aim to associate the dynamic effect of pedestrians with the fatigue limit state of pedestrian bridges. Future research should aim to continue exploring the proposed methodology with low-cost sensors. It can be achieved by comparing the results obtained by exploring other identification methodologies of dynamic parameters and between both types of sensors (low-cost and standard ones).

**Author Contributions:** Conceptualization, S.B., A.G. and A.O.; methodology, S.B.; software, S.B. and A.O.; validation, S.B., A.G. and A.O.; formal analysis, S.B.; investigation, S.B.; resources, S.B., A.G. and A.O.; data curation, S.B.; writing—original draft preparation, S.B.; writing—review and editing, S.B., A.G. and A.O.; visualization, S.B.; supervision, A.G and A.O.; project administration, A.G and A.O.; funding acquisition, S.B. All authors have read and agreed to the published version of the manuscript.

**Funding:** This research was funded by the following entities in Colombia: Sistema General de Regalías (SGR), the Ministerio de Ciencia, Tecnología e Innovación (CTeI) and the Universidad del Norte, through the call No. 1: "Convocatoria del fondo de ciencia, tecnología e innovación del sistema general de regalías para la conformación de una lista de proyectos elegibles para ser viabilizados, priorizados y aprobados por el OCAD en el marco del programa de becas de excelencia doctoral del bicentenario".

**Institutional Review Board Statement:** Not applicable.

**Informed Consent Statement:** Not applicable.

**Data Availability Statement:** The data presented in this study are available on request from the corresponding author.

**Acknowledgments:** The authors thank the Sistema General de Regalías (SGR), the Ministerio de Ciencia, Tecnología, e Innovación (CTeI) and the Universidad del Norte, whose through the call No. 1 of the CTeI of the Biennial Plan of Convocations in the framework of the bicentennial excellence scholarship program have supported this project; also thank to those who contributed to this research.

**Conflicts of Interest:** The authors declare no conflict of interest, and the funders had no role in the design of the study; in the collection, analyses, or interpretation of data; in the writing of the manuscript; or in the decision to publish the results.

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
