# Peer review of "Influence of Corrosion on Dynamic Behavior of Pedestrian Steel Bridges—Case Study"

_infrastructures, doi:10.3390/infrastructures8030052_

Round 1
Reviewer 1 Report
General Comment: The authors have attempted to establish a correlation between corrosion damage and dynamic parameters of 3 pedestrian bridges in a marine corrosion environment and thus leverage the dynamic properties of the bridge to estimate the degree of corrosion that the bridge may have undergone at a particular instant in their service life. While the methodology and approach are very interesting to achieve the objectives of this study, the results provided are not enough to establish the intended correlation and the lack of statistical analyses hamper the validity of the correlation that the authors have concluded in this study. While this paper is very interesting and the results have practical implications for pedestrian bridge inspection, it can only be accepted if the authors do more robust statistical analyses using a larger dataset to clearly and definitely support the conclusion of the existence of a correlation between the bridge dynamic parameters (seemingly only damping) and degree of corrosion in a bridge. The authors are strongly encouraged to conduct such analyses to get this manuscript to the acceptance level.
Some specific comments are provided as follows.
Specific Comments:
1. The first line of the abstract should be rewritten to clearly communicate the impact of corrosion on stiffness degradation in steel structures.
2. Line 32: Corrosion is a clear and visible threat in coastal structures and not a latent one. Please rephrase this statement.
3. The introduction section should be expanded and the existing techniques that are used for corrosion damage identification and quantification should be listed, particularly the low-cost technologies. Moreover, the authors should also provide some pros and cons of those methods and why they may not be ideal or optimum to adopt for the pedestrian bridges that trigger the need for this study.
4. A variety of AI-based methods are nowadays available that can be used to get a sense of corrosion level or corrosion damage in steel structures these days. Many of these methods use texture and pattern recognition to distinguish corrosion features and employ UAVs and other photo-capturing techniques. Since the premise of the paper is to envision a technique that can correlate the corrosion and dynamic behavior parameters, the authors should provide some background on those methods and justify the need for the technique the authors employed in this study.
5. The methodology is straightforward and does not need any modification.
6. Please describe the methodology used for estimating the corrosion damage and populating the corrosion matrix for each bridge. Is it based on qualitative or quantitative insights into the corrosion areas in the bridge? How much data was obtained to classify corrosion damage for each bridge?
7. The results are interesting, particularly in Fig. 16. However, the data provided in the preceding Table 5 shows that most of the dynamic parameters are relatively similar for all three bridges with the exception of damping. Damping of specific bridges has quite a spread in the data and hence it could be a statistical anomaly.
8. While the methodology and results are very interesting, the results need further statistical analysis to validate the existence and significance of a correlation between the dynamic parameters of the bridge and the degree of corrosion damage. However, the authors haven’t provided statistics to support the conclusion made in this study.
9. The correlation between the damping parameters and the degree of corrosion damage is based on three samples which are not enough to establish the correlation. The authors need to obtain more data points to establish a definite correlation between corrosion damage and damping.
Author Response
Dear reviewer, please see the attachment. After the letter of response to the reviewer's comments, you will find the new version of the manuscript.

Reviewer 2 Report
The study concerns the analysis of corrosion-induced effects on dynamic properties of steel truss pedestrian single-span bridges by using sensor-based modal identification. The investigated topic is of strong interest for current research trends in bridge management. The manuscript is well-written and clear. No corrections in English language are required. However, in this reviewer’s opinion. several modifications should be carried out to consider this study suitable for publication in the journal.
1. The introduction is clear. However, since the study concerns a particular bridge typology deserving conservation for its historical importance, the authors should consider adding some comments highlighting specific structural vulnerability issues for historical steel truss bridge. For example, issues on fatigue response (10.1016/j.engfailanal.2021.105996, 10.1016/j.engstruct.2010.10.008), robustness (10.1016/j.istruc.2020.12.005) and seismic response (10.1016/j.engstruct.2010.10.008) should be mentioned.
2. Additional information is required with reference to the numerical models: how the connection between the steel truss span and the abutmens is modelled? How the connection between diagonals and verticals to chords is modelled? Are non-structural loads included in the numerical model? These aspects are critical and may compromise the comparison between numerical model results and experimental outcomes.
3. Sectin 4 is strongly lacking in clarity. Extensive comments on the reported graphical and tabular outcomes are needed in Section 4.2 to 4.5.
a. It is not clear how the results of the numerical models are used to evaluate the influence of corrosion on bridge dynamic response. Are corrosion effects included in numerical models? How? According to table 5, it seems only a qualitative comparison is performed between numerical analysis and experimental results. If no model-updating processes are performed, numerical simulation is useless for the purpose of the study. The authors should clarify these aspects.
b. Figure 15 and Table 5 report three samples for experimental results. The authors should specify how and why three samples are collected.
c. How are box and whisker plots calculated? How is the variation is damping ratio and frequency computed?
d. The authors should explain why, in their opinion, the presence of corrosion is reflected by variations in experimental damping and/or frequency.
4. L316: The authors mention “There was no correlation between the natural frequencies of the bridge and the corrosion level measured.”. This is not true given the result reported in L309. Indeed, as well-known in literature, the frequency is affected by changes in stiffness induced by damage of structural component.
Author Response

(The authors gave the same response as above.)

Reviewer 3 Report
The objective of the Authors is very ambitious: try to detect corrosion damage in steel footbridges from modal testing under ambient conditions (operational modal analysis). The three considered case studies are indeed a valuable real-world laboratory to pursue that goal. However, the methodology adopted appears to be totally inappropriate. If this Reviewer has correctly understood, the Authors believe that a 5-minute vibration measuring campaign on each footbridge made in March 2022 should be able to provide general results on the possibility to correlate corrosion damage to measured dynamic properties. This is apparently a great jump that does not find sound scientific support. The collected data are largely insufficient to even try to think to such a correlation. Modal testing in general, and even so on completely exposed steel footbridges, provides results that are strongly influenced by environmental conditions (temperature, humidity, solar radiation, dominant speed direction and intensity) that should be filtered/compensated before any other correlation to possible damage can be even planned. In one footbridge the Authors also made comparisons with the results of previous vibration testing made in 2019 highlighting the differences. Again, are the Authors really sure that such differences are only correlated to corrosion and not to different environmental conditions? It is opinion of this Reviewer that the presented work is very immature and cannot be considered for publication. Indeed, the topic is really interesting and important for the engineering community involved in infrastructure safety evaluations and the case studies available to the Authors have great potentialities. However, the achievement of the research objective requires a much more in-depth study based on a much larger data collection over a sufficiently long monitoring period.
Author Response

(The authors gave the same response as above.)

Reviewer 4 Report
This manuscript studied the influence of corrosion in steel structures. Although, the topic and challenges mentioned in this work are interesting, there are several shortcomings that should be dealt with.
1) What is the main novelty of this work?
2) All subsections regarding the results of this manuscript contain short sentences with poor discussions and justifications.
3) Section 4.1: It seems that the result in Figure 11 is only based on engineering judgement. Is there any computational or analytical technique for achieving such result? More discussions are also needed for this section. Line 247: Figure 11 should be corrected to Figure 12.
4) Section 4.4: It seems that Table 4 presents the results of the first mode. How about the other modes? Are those identifiable?
5) What is the expression “sample” in Table 5?
6) Section 4.5: How did you ensure that the large variability in the damping ratios and slight increases in the natural frequencies were caused by corrosion? How about environmental and operational variability or different physical properties of the three bridges? The optimum way may be the comparison between the analytical and experimental model of any individual bridge, particularly after the analytical model updating.
7) Please use the expressions such as “section” or “sub-section” rather than “numerical”.
Author Response

(The authors gave the same response as above.)

Round 2
Reviewer 2 Report
The authors addressed the provided comments. This reviewer does not have further queries.
Reviewer 3 Report
The Authors have addressed the comments of this Reviewer. No further concerns.
Reviewer 4 Report
The authors properly revised the manuscript. I recommend publication.